# Cannabinoids and Their Receptors in Skin Diseases

**DOI:** 10.3390/ijms242216523

**Published:** 2023-11-20

**Authors:** Eun Hee Yoo, Ji Hyun Lee

**Affiliations:** Department of Dermatology, Seoul St. Mary’s Hospital, College of Medicine, Catholic University of Korea, Seoul 06591, Republic of Korea; eunheeryoo0312@gmail.com

**Keywords:** cannabinoids, cannabidiol (CBD), cannabinoid receptor 1 (CB1R), cannabinoid receptor 2 (CB2R), endocannabinoids, eczema, atopic dermatitis, psoriasis, acne, hair growth, skin aging

## Abstract

The therapeutic application of cannabinoids has gained traction in recent years. Cannabinoids interact with the human endocannabinoid system in the skin. A large body of research indicates that cannabinoids could hold promise for the treatment of eczema, psoriasis, acne, pruritus, hair disorders, and skin cancer. However, most of the available data are at the preclinical stage. Comprehensive, large-scale, randomized, controlled clinical trials have not yet been fully conducted. In this article, we describe new findings in cannabinoid research and point out promising future research areas.

## 1. Introduction

In recent years, some components of cannabis, also known as marijuana, have been studied. Cannabis has been used for various purposes throughout history, including recreational, medicinal, and industrial uses. In recent years, cannabinoid components are emerging as therapeutic alternatives for patients with a variety of illnesses and conditions. In particular, their anti-inflammatory properties have piqued the interest of dermatologists [1]. Given the growing number of pre-clinical and clinical studies exploring the potential of cannabinoids to treat dermatologic conditions, we here summarize reports of cannabinoid use in dermatologic therapy.

### 1.1. Types of Cannabinoids

Cannabinoids are a significant group of biologically active substances similar to the primary psychoactive compound derived from *Cannabis sativa* [2,3]. They are divided into three classes based on their site of production (Table 1). Endocannabinoids are naturally produced in human bodies. Phytocannabinoids are plant-derived cannabinoids that occur in *Cannabis* plants, and synthetic cannabinoids are produced in laboratories through chemical processes. Cannabis contains more than 140 phytocannabinoids, including cannabidiol (CBD) and Δ-9-tetrahydrocannabinol (THC), which differ in structure slightly, and endocannabinoid-like lipid mediators such as palmitoylethanolamide (PEA) [4].

### 1.2. Endocannabinoid System in Human Skin

The endocannabinoid system consists of endocannabinoids, receptors, and enzymes responsible for the synthesis or degradation of endocannabinoids, and it functions as a molecular signaling system. The endocannabinoid system is shown in Figure 1. The endocannabinoid system plays an important role in skin homeostasis, and its dysregulation has been linked to dermatologic diseases [5].

**Table 1 ijms-24-16523-t001:** Cannabinoid classes and abbreviations.

Types of Cannabinoids	Constituents
Endocannabinoids (ECB)	Palmitoylethanolamide (PEA)
	Anandamide (AEA) or N-arachidonoylethanolamine
	2-arachidonoyl-glycerol (2-AG)
Phytocannabinoids	Δ(9)-Tetrahydrocannabinol (THC)
	Cannabidiol (CBD)
	Cannabigerol (CBG)
	Cannabichromene (CBC)
	Cannabigerovarin (CBGV)
	Cannabinol (CBN)
	Cannabigerolic acid (CBGA)
Synthetic cannabinoids	WIN55,212-2
	JTE-907

The most well-known endocannabinoids are 2-arachidonoyl-glycerol (2-AG) and anandamide (AEA). They are derived from cell membrane phospholipids and are the natural ligands for their receptors, cannabinoid receptor 1 (CB1R) and cannabinoid receptor 2 (CB2R) [6]. CB1R and CB2R are G-protein coupled receptors that comprise the endocannabinoid system. CB1R is typically found in high concentrations in the central nervous system, whereas CB2R is predominantly found in the peripheral nervous system. Previous studies suggest that both CB1R and CB2R receptors are present in keratinocytes, mast cells, hair follicles, and sensory nerve fibers in the skin. In epidermal keratinocytes, hair follicles, and sebaceous glands, CB1R and CB2R receptors show a complementary distribution pattern [7,8,9].

### 1.3. Other Receptors Activated by CANNABINOIDS

Though endocannabinoids primarily bind to cannabinoid receptors, they are also known to bind to transient receptor potential (TRP) receptors [10]. TRP channels are a group of transmembrane ion channels engaged in various physiological functions, such as the perception of itch, temperature, and pain [11]. AEA is the first known endogenous agonist of TRPV1, and THC, a major phytocannabinoid, interacts most potently with TRPV2. There is increasing evidence that cannabinoids interact with TRP receptors, suggesting that targeting TRP may also hold promise for dermatological treatments [12].

Peroxisome proliferator-activated receptor (PPAR) is another receptor which interact with cannabinoids by both direct and indirect pathways. PPAR is known to regulate lipid metabolism, glucose metabolism, and inflammation. PEA is well known agonist of PPAR-α [2,6].

## 2. Eczema, Atopic Dermatitis, Allergic Contact Dermatitis 

### 2.1. Allergic Contact Dermatitis 

Allergic contact dermatitis is a T-cell-mediated delayed-type hypersensitivity occurring after exposure to an allergen. Recently, cannabinoids and their receptors were shown to be associated with allergic contact dermatitis.

#### 2.1.1. Cannabidiol (CBD) and Allergic Contact Dermatitis 

The anti-inflammatory effect of CBD is demonstrated in several experimental studies and clinical studies. In an in vitro model of allergic contact dermatitis, CBD demonstrated dose-dependent inhibition of monocyte chemotactic protein 2, which attracts mast cells and macrophages to inflammatory sites, and other pro-inflammatory cytokines such as IL-6, IL-8, and TNF-α. This anti-inflammatory effect of CBD was reversed by a selective CB2R antagonist and TRPV1 antagonist [13] (Figure 2).

Several clinical studies have shed light on the potential benefits of CBD and CBD-containing hemp oil in eczematous dermatitis. An observational study demonstrated that the topical application of CBD gel reduced a patient-oriented eczema measurement score from 16 to 8.1 (*p* < 0.001) [14]. Also, in patients with atopic dermatitis, the topical application of CBD ointment significantly improved transepidermal water loss (TEWL) (*p* < 0.001) and clinical severity [15]. A randomized, single-blinded, crossover study of 20 atopic dermatitis patients found that dietary hemp oil increased polyunsaturated fatty acid levels, which improved skin dryness and itching [16].

#### 2.1.2. Δ-9-Tetrahydrocannabinol (THC) and Allergic Contact Dermatitis 

In vitro experiments demonstrated that THC decreased keratinocyte-derived pro-inflammatory mediators such as CCL2, CCL8, and CXCL10, which are induced by IFN-γ, limiting myeloid immune cell infiltration [17] (Figure 2).

The anti-inflammatory effect of THC was also studied in animal models. In a 2,4-dinitrofluorobenzene (DNFB)-induced allergic contact dermatitis mouse model, the topical application of THC attenuated allergic inflammation. In that study, mice lacking both CB1R and CB2R displayed aggravated ear swelling [18]. As studies on THC are scarce, more preclinical and clinical studies are needed to confirm the anti-inflammatory effects of THC.

#### 2.1.3. CB1R/CB2R and Allergic Contact Dermatitis 

The relation between endocannabinoid receptor and allergic contact dermatitis is also demonstrated in several studies. The skin irritant polyyne falcarinol, a CB1R receptor antagonist, increased the expression of the pro-allergic chemokines IL-8 and CCL2/MCP-1 in vitro and aggravated histamine-induced wheals in a skin prick test in vivo, which imply an association between CB1R and allergic dermatitis [19] (Figure 2).

The association between CB2R and cutaneous inflammation varied. S. Oka et al. found that a CB2R agonist suppressed inflammatory reactions in mouse ears induced by 12-O-tetradecanoylphorbol 13-acetate [20]. However, in other studies, the topical application of the CB2R agonist HU-308 either had no effect or increased allergic reaction [18], and a CB2R antagonist or inverse agonist suppressed allergic inflammation in mice [20,21]. The conflicting results suggest that the effects of CB2R antagonism could be beneficial or harmful depending on the concentration or duration.

### 2.2. Atopic Dermatitis 

As with allergic contact dermatitis, the link between cannabinoids and atopic dermatitis has been demonstrated in numerous studies. Because the endocannabinoid system is associated with the pathophysiology of atopic dermatitis, treatments targeting the endocannabinoid system are gaining traction.

#### 2.2.1. CB1R/CB2R and Atopic Dermatitis

The role of CB1R/CB2R has mostly been elucidated in in vivo studies. In the case of CB1R, a CB1R agonist has been shown to reduce inflammations in an atopic dermatitis model. The topical application of a CB1R-specific agonist, α-oleoyl oleylamine serinol, significantly facilitated the recovery of the epidermal permeability barrier (*p* < 0.01) and prevented epidermal thickening in an atopic dermatitis model [22]. Similarly, mice lacking CB1R showed enhanced ear swelling and delayed recovery of the epidermal barrier in an atopic dermatitis model, which is consistent with previous studies. The research showed that the pathogenesis was due to increased levels of IL-4, thymic stromal lymphopoietin (TSLP), IFN-γ, and CCL8, pro-inflammatory molecules which drive Th2-type inflammation in atopic dermatitis [23] (Figure 2).

GW Nam et al. demonstrated that the topical application of selective CB1R agonists suppressed mast cell proliferation in vitro and mast cell infiltration in oxazolone-induced skin in vivo. Those findings indicate the potential usefulness of CB1R agonists in alleviating inflammatory symptoms triggered by mast cell activation, such as that seen in atopic dermatitis and contact dermatitis [24]. Unlike CB1R agonists, little research has considered CB1R antagonists, and further research is needed to provide more clarity on the role of CB1R in atopic dermatitis.

Several studies have also considered the role of CB2R in atopic dermatitis. JTE-907, a selective CB2R antagonist, suppressed scratching behavior and cutaneous nerve activity in NC mice, a model of atopic dermatitis. Also, oral administration of JTE-907 (10 mg/kg) has been shown to improve the dermatitis score to an extent similar to tacrolimus [25]. As we described above, CB1R knockout mice showed delayed epidermal barrier recovery. However, CB2R-deficient mice showed enhanced epidermal permeability recovery [26]. The CB2R results are as mixed for atopic dermatitis as they are for allergic contact dermatitis, so further research is needed.

#### 2.2.2. Palmitoylethanolamide (PEA) and Atopic Dermatitis 

PEA is an endocannabinoid-like lipid mediator that functions primarily through activating nuclear receptor peroxisome proliferator-activated receptor-α (PPAR-α) and working through the endocannabinoid system [27]. PEA is known to bind CB2R on mast cells and regulate immune responses via mast cell stabilization and Th2 cell downregulation in atopic dermatitis, which is the likely reason for its anti-pruritic, anti-inflammatory effects [28].

The effect of PEA in atopic dermatitis has been relatively well studied in clinical trials. A randomized controlled trial involving 60 patients with asteatotic eczema demonstrated that emollient creams containing PEA or AEA exhibited greater efficacy than the control in improving scaling, dryness, and itching (*p* < 0.05). Also, skin surface hydration was improved in the PEA/AEA emollient group compared with the control [29]. In a split-body clinical study of 43 patients with atopic dermatitis, the body sides treated with the combination of a topical steroid and PEA showed more rapid clearance than the sides treated with only a topical steroid [30]. In addition, in a prospective cohort study of 2456 atopic dermatitis patients, the topical application of PEA produced substantial relief from objective symptoms such as dryness, excoriation, lichenification, scaling, erythema, and pruritis in 58.6% of the patients [31]. Topical adelmidrol, an analog of PEA, has also been shown to be effective in treating mild atopic dermatitis in a pediatric population. After four weeks of treatment, the complete clearing of the lesions was observed in 16 (80%) patients [32]. Considering the importance of minimizing the use of topical corticosteroids when treating eczema due to safety issues and side effects, the anti-inflammatory and anti-pruritic effects of PEA make it a promising treatment for atopic dermatitis and allergic contact dermatitis. However, its molecular pathogenesis should be clarified.

## 3. Pruritis

### 3.1. Pruritis and Cannabinoid Receptors 

Cannabinoids and their receptors have been identified as neuronal modulators of the pruritic response, making them a potential target for the itch associated with various dermatologic diseases.

Several in vivo studies investigated the antipruritic effect of endocannabinoid system.

X Liu et al. demonstrated that mice with CB1R knockout in their primary sensory neurons showed aggravated inflammation and itch. CB1R knockout mice in primary sensory neurons promoted the production of substance P, which affected the activation of extracellular signal-regulated kinase (ERK) in keratinocytes. This pathway induced the accumulation of mast cells in the dermis (Figure 3) [33]. Similarly, CB2R knockout mice also showed exacerbated psoriasiform dermatitis and itching compared with wild-type mice [34].

### 3.2. Pruritis and Cannabinoid Receptor Agonist 

A CB1R/CB2R agonist, WIN 55,212-2, ameliorated chronic itching through suppressing Il-13 and IL-31 in mice (Figure 3) [35]. Interestingly, other studies found that the anti-pruritic effects of WIN 55,212-2 are partially mediated by spinal CB1R [36].

A selective CB2R agonist, S-777469, significantly inhibited scratching compared with fexofenadine in an in vivo animal model [37]. In twelve healthy males, the peripheral administration of a CB1R/CB2R agonist (HU210) reduced histamine-induced itch [38].

### 3.3. Pruritis and Transient Receptor Potential (TRP) Ion Channels

TRP ion channels are abundant in the skin and nervous system, where they mediate sensory responses such as itching. To date, six TRP channels—TRPV1, TRPV2, TRPV3, TRPV4, TRPA1, and TRPM8—are known to be involved in both endogenous and exogenous cannabinoid signaling [39]. TRPV1, recognized as the capsaicin receptor, has pruritic properties. TRPV1 is increased in atopic dermatitis-like skin lesions and subsequently released substance P and neurokinin-1 receptor. And TRPV1 inhibition consistently reduces pruritis [40,41].

Many clinical studies about cannabinoid treatment of pruritis have used PEA, which causes desensitization to TRPV1 [42]. In an observational, non-blinded, prospective cohort study, three weeks of topical treatment with PEA and anandamide significantly reduced both pruritis and scales in 38% of 21 uremic patients (*p* < 0.0001) [43,44]. Also, cream containing PEA decreased pruritis associated with atopic dermatitis, lichen simplex, and prurigo nodularis in 14 of 22 patients [45]. However, the PEA-containing lotion did not differ significantly from the control group in a randomized, single-blinded study of 100 patients with chronic pruritis [46]. Additional clinical trials for PEA are thus needed to confirm its antipruritic effect.

### 3.4. Pruritis and Δ-9-tetrahydrocannabinol (THC) Ion Channels

THC has also been reported to be an effective treatment for itching in several studies. In case reports, patients with epidermolysis bullosa who were treated sublingually with CBD and THC reported improved pain scores and pruritis [47]. The oral administration of 5 mg of THC produced marked improvement in pruritis and quality of life in patients with intractable cholestatic-related pruritus [48].

### 3.5. Pruritis and Fatty Acid Amide Hydrolase (FAAH)

Fatty acid amide hydrolase (FAAH), the primary enzyme responsible for degrading the endocannabinoid anandamide (AEA), is a novel target for the treatment of pruritis. The role of FAAH is well studied in in vivo models.

FAAH knockout mice and mice treated with FAAH inhibitors, which all had elevated levels of fatty acid amides, demonstrated significantly less scratching behavior than wild-type mice. Interestingly, mice expressing FAAH exclusively in neuronal tissue exhibited scratching responses similar to those of wild-type mice, suggesting that FAAH inhibition in the nervous system is essential to the antipruritic effect. Also, the antipruritic effect of FAAH deficiency was reversed by a CB1R antagonist [49]. The systemic and spinal administration of FFAH produced dose-dependent antipruritic effects in a serotonin-induced scratching model [50,51]. Because CBD is an FFAH inhibitor, it could potentially play a role in modulating itch responses, but the scientific evidence for that effect is inadequate.

## 4. Psoriasis

Cannabinoids might improve psoriasis through reversing keratinocyte inhibition and inhibiting the release of inflammatory cytokines such as IL-2, TNF-α, and IFN-γ, which are important in the pathogenesis of psoriasis [52].

### 4.1. Psoriasis and Endocannbinoids

Wilkinson JD et al. demonstrated that CBD, THC, cannabinol, and cannabigerol (CBG) dose-dependently inhibited keratinocyte proliferation in a way that was independent of the cannabinoid receptor. Those authors hypothesized that the main mechanism could be an interaction between the cannabinoids and PPAR-γ (Figure 4) [53]. The CB1R agonist AEA has been shown to suppress the proliferation of human keratinocytes through its action on CB1R and its activation of TRPV1-mediated calcium influx [54].

NF-κB is a key factor in various dermatologic diseases including psoriasis, and NF- κB activation is induced by TNF- α. Sangiobanni et al. showed that CBD and *C. sativa* extract inhibited TNF-α-induced NF-κB transcription in human keratinocyte cells. In this study, *C. sativa* extract downregulated IL-8, which regulates keratinocyte proliferation, and VEGF, which plays an important role in angiogenesis and psoriasis. In addition, CBD showed a concentration-dependent inhibition of MMP9, which degrades the extracellular matrix (Figure 4) [55].

### 4.2. Psoriasis and Cannabinoid Receptors

Several findings suggest that both CB1R and CB2R play a pivotal role in the pathophysiology of psoriasis. Psoriasis is characterized by the upregulation of keratin K6 and K16 expression within the epidermis [56]. Ramot Y. et al. found that a CB1R-specific agonist, arachidonoyl-chloro-ethanolamide, decreased K6 and K16 expression in keratinocytes in situ [57].

Li et al. found that CB2R-deficient mice showed exacerbated imiquimod-induced psoriasiform dermatitis and itching compared with wild-type mice. CB2R deficiency upregulated the expression of proinflammatory cytokines such as TNF-α, IL-1β, and IL-17 through increasing the infiltration of CD4^+^ T cells and the Th17/Treg ratio. Those effects were reversed by the CB2R agonist JWH-133 [34]. Also, CB2R deficiency inhibited the differentiation of CD4 T cells through ROR γ T [34]. CB2R activation reduced IL-17 production by Th17 lymphocytes via a STAT5 dependent pathway (Figure 4) [58]. Another study demonstrated that human mesenchymal stem cells express all of the components of the endocannabinoid system, including CB1R/CB2R. In mesenchymal stem cells, CB2R stimulation reduced pro-inflammatory cytokines such as IL-1β, IL-8, and IL-6 [59].

Similarly, mice with CB1R knockout in their primary sensory neurons showed aggravated imiquimod-induced psoriasiform inflammation and itch. Also, inhibiting substance P, which induces mast cell accumulation and activation, reversed the aggravation of pruritis and inflammation in CB1R knockout mice [33].

### 4.3. Psoriasis and CBD in Clinical Trials

The effect of CBD on psoriasis has also been investigated in clinical trials. Friedman et al. demonstrated that psoriasis patients treated daily with a topical THC cream showed improvement within two days [60]. Also, five patients with psoriasis who applied topical CBD-enriched ointment on their lesions twice daily for three months had an overall improvement in their Psoriasis Severity Index scores on day 90 (*p* < 0.001) [15]. Patients with scalp psoriasis who were treated with CBD oil experienced reduced arborizing vessels and inflammation. They also reported less itching and burning symptoms after using the CBD oil [61]. A split-body, double-blind, placebo-controlled study of 51 patients with mild plaque-type psoriasis was conducted. The patients applied 2.5% CBD ointment and placebo twice daily for 12 weeks and showed significantly lower Psoriasis Severity Index scores on the CBD side throughout the follow-up period (*p* = 0.026) [62].

## 5. Acne and Seborrhea

Endocannabinoids show promise as a potential therapeutic agent for the treatment of acne and seborrhea. It is known that both CB1R and CB2R are expressed in human sebaceous glands [8]. This suggests that cannabinoids may influence lipogenesis and inflammatory responses in sebocytes.

### 5.1. Cannabinoids and Inflammation in Sebocytes

A previous study showed that the endocannabinoids AEA and 2-AG induced lipid production dose-dependently in human sebocytes, and that production was mediated by CB2R-coupled signaling (not CB1R) in the MAPK pathway. Also, they showed that AEA and 2-AG stimulated apoptosis in a CB2R-dependent manner. These data suggest that human sebocytes are both sources and targets of endocannabinoids (Figure 5) [63].

In contrast, Olah et al. found that different cannabinoids produced different responses in sebocytes: the non-psychotropic phytocannabinoids (−)-cannabichromene and (−)-Δ^9^-tetrahydrocannabivarin suppressed AEA-induced lipogenesis and arachidonic-induced seborrhea-mimicking lipogenesis, but CBG and (−)-cannabigerovarin increased it. All tested phytocannabinoids showed anti-inflammatory actions in sebocytes, indicating their potential as anti-acne agents [64].

Because the overgrowth of *Cutibacterium acnes* (*C. acnes*) has been linked to acne, the anti-microbial properties of CBD might also be effective in treating acne. CBD suppressed lipogenesis in sebocytes, and these sebostatic actions of CBD are mediated by the TRPV4 ion channel. Also, they found that the downstream signaling pathway is the ERK 1/2 MAPK pathway, which is a prolipogenic signaling pathway in sebocytes (Figure 5) [65].

A hemp seed hexane extract inhibited the IL-1β and IL-8 induced by *Propionibacterium acnes* and showed inhibitory effects on lipid production through the AMPK and AKT/FoxO1 signaling pathways. Also, hemp seed hexane extract inhibited the NF-κB and MAPK pathway in keratinocytes [8,66]. Jian et al. demonstrated that CBD inhibited the inflammation induced by *C. acnes*–derived extracellular vesicles in human keratinocytes, which is mediated by CB2R. They found that CBD upregulated CB2R expression and downregulated TRPV1 expression, which resulted in the inactivation of the MAPK and NF-κB signaling pathways in keratinocytes (Figure 5) [67].

### 5.2. Cannabinoids in Acne and Seborrhea—Clinical Studies 

Clinical studies are still scarce. In a single-blinded study of 11 human patients, 3% cannabis-seed-extract cream applied twice a day for 12 weeks produced significant improvement in sebum production and erythema, measured using a reflectance spectrophotometer (Mexameter) (*p* < 0.05) [68]. A phase 2 clinical trial of a 5% CBD topical for acne (BTX1503) is underway, but results have not yet been published [69]. Although no large-scale human trials have been carried out, we speculate that hemp seed and cannabinoids could be used for acne treatment because of their anti-lipogenic, anti-inflammatory, and anti-microbial effects.

## 6. Hair Growth 

The human hair follicle is an immune-privileged miniaturized organ of epithelial and mesenchymal tissue. The process of hair growth consists of a period of keratinocyte proliferation and hair fiber growth (anagen), followed by apoptotic follicle regression (catagen) and a semi-quiescent stage (telogen) [70].

Immunohistochemistry examinations of human hair follicles have shown the differential distribution of CB1R and CB2R receptors. CB1R was detected in the infundibulum and the inner hair root sheath, but not in the outer root sheath, bulge, bulb of the hair follicles, or the arrector pili muscle. In contrast, CB2R was found in the undifferentiated cells of the infundibulum, the outer hair root sheath, and the bulb of hair follicles [8].

### 6.1. Cannabinoids Receptor 1 (CB1R) and Hair Growth 

Both AEA and THC dose-dependently inhibited hair shaft elongation and the proliferation of hair matrix keratinocytes. They also induced intraepithelial apoptosis and premature hair follicle regression. A selective antagonist of CB1R reversed those effects [71]. Also, a synthetic CB1R antagonist, thienyl substituted pyrazole carboxamide derivative, stimulated hair growth in diet-induced obese mice with body weight reduction. However, no such hair growth stimulation was observed after a topical application of the antagonist [72]. Clinically, these data suggest that CB1R antagonists might counteract hair loss. The effect of CB1R antagonist on hair loss should be further elucidated in clinical studies.

### 6.2. Cannabidiol (CBD) and Hair Growth 

In a pilot study using ex vivo human hair follicles and primary outer root sheath keratinocytes, CBD showed variable effects on both the elongation and hair cycle of human hair follicles. In that study, the researchers proposed that CBD concentration could lead to differential receptor activation, with a low dose (0.1 μM) favorably affecting hair growth pathways and a higher dose (10 μM) reducing hair shaft development through activating TRPV4 [73].

In a case series, 35 patients with androgenetic alopecia used a once-daily topical hemp oil formulation, averaging 3–4 mg of CBD per day, and reported significant hair growth after six months with no side effects [74]. These findings all suggest that cannabinoids, which interact with TRPV channels, could function as a promising target for human hair growth control.

### 6.3. Bimatoprost and Hair Growth 

Bimatoprost is a topical prostamide that is a metabolite of the endocannabinoid AEA and is known to be effective in eyebrow hypotrichosis [75]. Khidhir et al. demonstrated that bimatoprost at specific concentrations stimulated the growth of hair follicles in a follicle organ culture and mice in vivo [76]. In a randomized controlled trial of 30 patients with alopecia areata, two scalp alopecia areata in one patient were randomly treated with either topical corticosteroid cream or bimatoprost solution. The patches treated with topical bimatoprost showed a significantly earlier response, faster hair re-growth, and a lower incidence of resistance and relapses than the corticosteroid-treated patches [77].

Thus, existing studies show that endocannabinoids have a potential role in treating alopecia. However, CBD can cause hair growth, and several other phytocannabinoids can lead to hair loss. Therefore, given the complexity of hair-growth dynamics, further research, including clinical trials, is needed [78].

## 7. Skin Cancer 

### 7.1. Cannabinoids and Melanoma 

Cannabinoids might also be useful in managing both melanoma and non-melanoma skin cancer [79]. Blazquez et al. demonstrated that both CB1R and CB2R are expressed in murine and human melanoma cells. In vitro experiments on the A353 and MelJuso melanoma cell lines demonstrated that cannabinoids significantly decreased the number of viable melanoma cells through inducing apoptosis in the cultures. Also, those effects were reversed using CB1R and CB2R antagonists.

Interestingly, the anti-melanoma effect was selective for melanoma cells and did not affect normal melanocytes, even though they also expressed CB1R. In addition, CB2R agonists inhibited melanoma progression and metastatic spread in mice [80]. Local administration of the mixed CB1R/CB2R agonist WIN-55,212-2 or the selective CB2R agonist JWH-133 induced considerable growth inhibition of malignant tumors in mice. Cannabinoid-treated tumors showed an increased number of apoptotic cells, accompanied by decreased expression of VEGF [81]. Mice with melanoma that were treated with an intraperitoneal injection of CBD 5 mg/kg twice per week showed a significant decrease in tumor size and a better survival curve than the control group. Mice treated with cisplatin demonstrated the longest survival time, but the quality of life and movement of the CBD-treated mice were observed to be better [82].

THC has also been shown to have anti-cancer properties. Treatment with THC resulted in the activation of autophagy, apoptosis, and a loss of cell viability. Administration of a THC and CBD mixture to mice with melanoma xenografts significantly inhibited melanoma viability, proliferation, and tumor growth, paralleled by an increase in apoptosis, compared with the standard treatment of temozolomide [83].

### 7.2. Cannabinoids and Kaposi Sarcoma

Cannabinoids can also potentially influence Kaposi sarcoma. The CB1R/CB2R agonist WIN-55,212-2 showed anti-mitogenic effects on Kaposi sarcoma cells [84]. Also, CBD reduced proliferation and induced apoptosis in endothelial cells infected by Kaposi sarcoma-associated herpesvirus. CBD inhibited the expression of viral G-protein coupled receptor and VEGF [85]. However, the efficiency of Kaposi sarcoma-associated herpesvirus infection was increased in the presence of low-dose THC, and THC stimulated the expression of the viral G-protein coupled receptor and the proliferation of Kaposi’s sarcoma cells [86].

These findings indicate that cannabinoids hold great promise as cytotoxic agents in the treatment of skin cancer in specific doses and should be further evaluated in clinical trials.

## 8. Skin Aging 

Cannabinoids have also demonstrated a role in skin rejuvenation and anti-aging.

### 8.1. Cannabinoids Receptor (CB1R) and Anti-Aging 

The relationship between CB1R and skin aging has been well studied in vivo. Compared with control mice, mice with genetic deletion of CB1R showed a thinner subdermal fat layer, which is an aging-like histological change in the skin [87]. In another study, mice with CB1R deficiency showed decreased collagen production and higher expression of pro-inflammatory markers than control mice [88]. Those findings suggest that the CB1R receptor is associated with the aging process in skin.

### 8.2. CBD and Anti-Aging 

The anti-aging properties of CBD have been demonstrated in several studies so far. Both THC and CBD in the range of 0.5 µM to 2.0 µM stimulated cell growth in a dose-dependent manner while significantly decreasing senescence, as measured by beta-galactosidase activity, in human dermal fibroblasts. In a scratch assay, both THC and CBD (2.0 µM) significantly improved wound healing in both healthy and stress-induced premature senescent skin fibroblasts, performing better than metformin, rapamycin, and triacetyl resveratrol [89]. Also, flax fiber, which contains CBD, inhibited inflammation and activated skin cell matrix remodeling. Those authors postulated that the anti-inflammatory and collagen production effects were mainly due to the CBD content [90]. CBD induced the expression of heme-oxygenase 1 and increased the levels of proliferation and wound-repair-associated keratins 16 and 17 in the skin of mice, which is a possible mechanism for the anti-aging effects of CBD [91].

Therefore, cannabinoids can be valuable sources for skin rejuvenation and anti-aging in cosmetics, and more studies, including clinical trials, should be performed.

## 9. Conclusions

Preliminary studies indicate that cannabinoids can improve eczema, acne, pruritus, psoriasis, hair growth, and skin cancer and have anti-aging effects on skin. However, the pharmacology of cannabinoids in controlling skin diseases is complex and not yet completely understood.

There are several clinical trials that show the potential benefits of cannabinoids in various dermatologic diseases (Table 2). Notably, cannabinoids have recently expanded their presence in the market, shifting from cosmetics that alleviate skin inflammation to encompass anti-aging and hair treatment products. Clinical trials involving cannabinoids for pain, inflammation, and cancer are actively ongoing, with a growing interest in their potential application in conditions that extend beyond skin concerns.

It is worth noting, however, that European Union cosmetics regulations stipulate that all natural raw materials derived from hemp in cosmetics must originate from parts of the Cannabis sativa plant, with the added condition that the total THC content should not exceed 0.2%. Currently, there is a movement in Korea to relax regulations concerning the use of cosmetics, albeit with the provision that only CBD with low THC content is permitted. To navigate this evolving landscape, individuals and industries involved in research and cosmetics must familiarize themselves with the specific regulations of each country. The potential for the use of cannabinoids in various sectors appears to be limitless, presenting a wide array of opportunities.

## Figures and Tables

**Figure 1 ijms-24-16523-f001:**
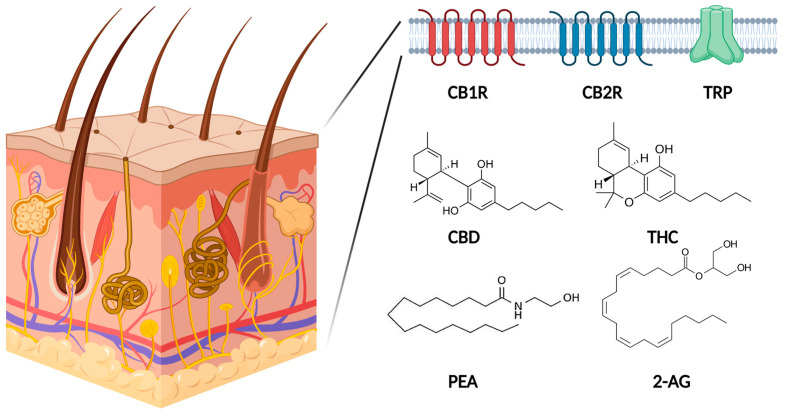
Schematic presentation of the endocannabinoid system. The endocannabinoid system consists of endocannabinoids, receptors, and enzymes. Cannabidiol (CBD) and Δ-9-tetrahydrocannabinol (THC) are phytocannabinoids, and palmitoylethanolamide (PEA) and 2-arachidonoyl-glycerol (2-AG) are endocannabinoids. CB1R, cannabinoid receptor 1; CB2R, cannabinoid receptor 2; TRP, transient receptor potential; CBD, cannabidiol; THC, Δ-9-tetrahydrocannabinol; PEA, palmitoylethanolamide; 2-AG, 2-arachidonoyl-glycerol. This figure was created with Biorender at www.biorender.com (accessed on 21 August 2023).

**Figure 2 ijms-24-16523-f002:**
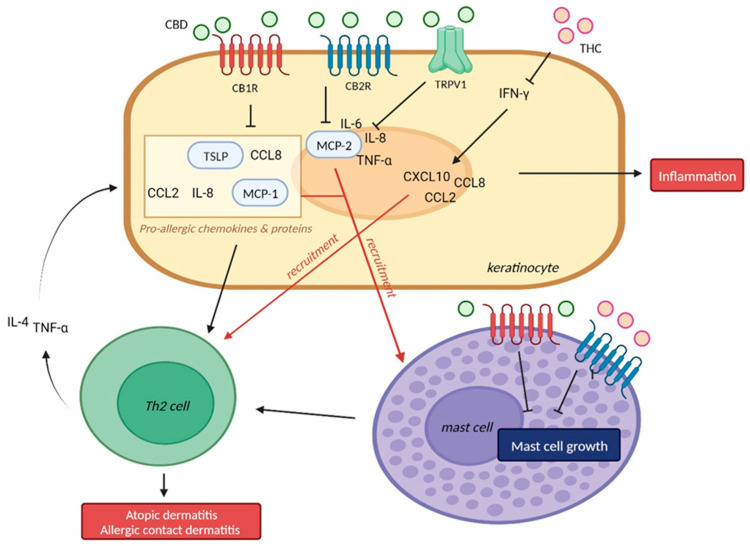
Mechanism of action of CBD and THC in atopic dermatitis and eczema. CBD inhibited pro-inflammatory cytokines such as IL-6, IL-8, and TNF-α. CB1R-deficient mice showed increased levels of IL-4, TSLP, IFN-γ, and CCL-8, which induces Th2-type inflammation. The interaction between cannabinoids and CB1R/CB2R suppressed cytokines including MCP-1, MCP-2, which recruits mast cell. Mast cells are also known to express CB1R/CB2R, which inhibited mast cell growth. THC decreased keratinocyte-derived pro-inflammatory mediators such as CCL2, CCL8, and CXCL10, which are induced by IFN-γ, limiting myeloid immune cell infiltration. TSLP, thymic stromal lymphopoietin; MCP, monocyte chemoattractant protein; Th2, T-helper 2 cell; TNF-α, tumor necrosis factor; CBD, cannabidiol; THC, Δ-9-tetrahydrocannabinol; TRPV, transient receptor potential vanilloid. This figure was created with Biorender at www.biorender.com (accessed on 21 August 2023).

**Figure 3 ijms-24-16523-f003:**
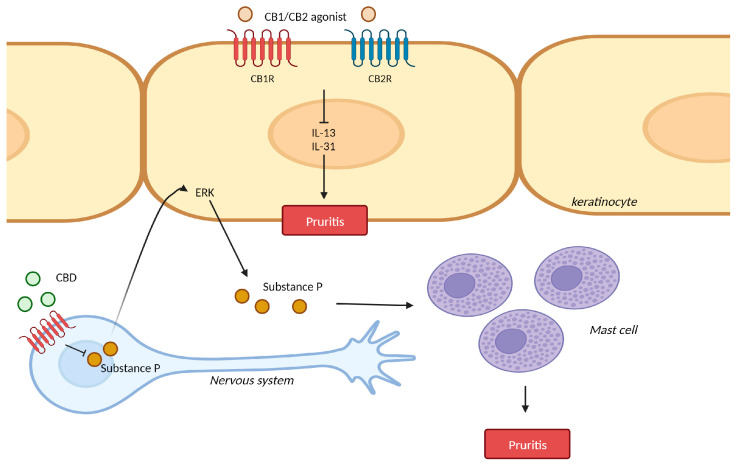
Mechanism of action for cannabinoids in pruritis. CB1R/CB2R agonist ameliorated chronic itching by suppressing Il-13 and IL-31. CB1R knockout in sensory neurons promoted the production of substance P, which induces the accumulation of mast cells in the dermis through ERK. CBD: cannabidiol; ERK: extracellular signal-regulated kinase. This figure was created with Biorender at www.biorender.com (accessed on 21 August 2023).

**Figure 4 ijms-24-16523-f004:**
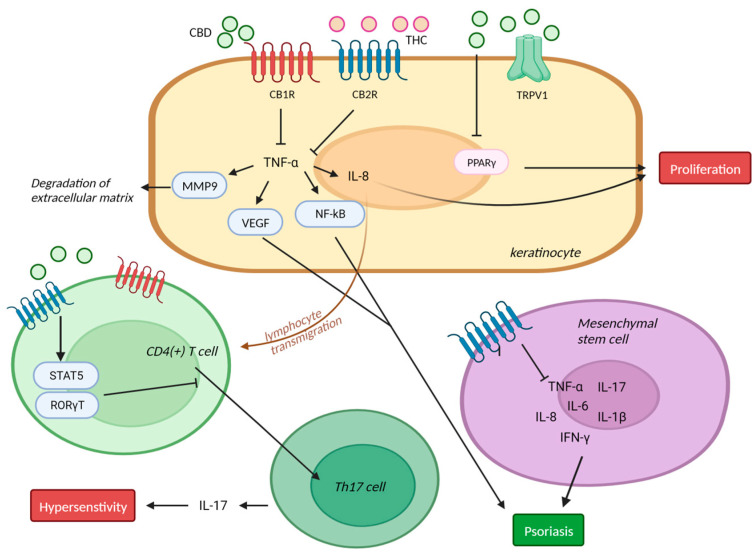
Mechanism of action of cannabinoids in psoriasis CBD inhibited TNF-α-induced NF-κB transcription, which plays important role in psoriasis, and keratinocyte proliferation through PPAR-γ. CB2R deficiency inhibited the differentiation of CD4 T cells through STAT5/ROR γ T. Human mesenchymal stem cells express all of the components of endocannabinoid system, including CB1R/CB2R. And CB2R stimulation reduced pro-inflammatory cytokines in mesenchymal stem cells. MMP: matrix metallopeptidase; Th17: T-helper 17 cell; TNF-α: tumor necrosis factor; CBD: cannabidiol; THC: Δ-9-tetrahydrocannabinol; TRPV: transient receptor potential vanilloid; NF-κB: nuclear factor kappa-light-chain-enhancer of activated B cells; PPAR-γ: peroxisome proliferator-activated receptor; ROR: AR-related orphan nuclear receptor. This figure was created with Biorender at www.biorender.com (accessed on 21 August 2023).

**Figure 5 ijms-24-16523-f005:**
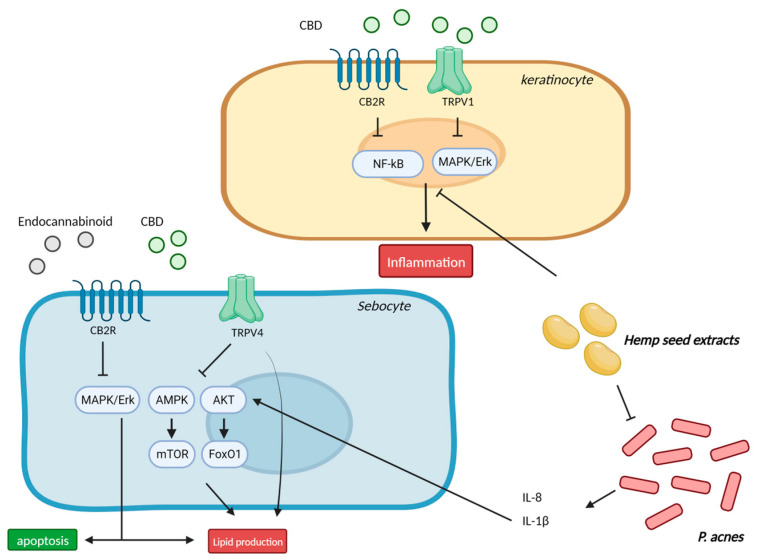
Mechanism of action of cannabinoids in acne and seborrhea Endocannabinoids enhance lipid synthesis and apoptosis of sebocytes through CB2R dependent manner. The lipostatic action of CBD is mediated by the TRPV4 ion channel through the inhibition of the MAPK/Erk pathway. A hemp seed hexane extract inhibited the NF- κB and MAPK pathway in keratinocytes and regulated lipid production via AMPK and AKT/FoxO1 signaling in sebocytes. CBD: cannabidiol; TRPV: transient receptor potential vanilloid; MAPK/Erk: mitogen-activated protein kinase/extracellular signal-regulated kinase 1/2; AMPK: AMP-activated protein kinase; mTOR: mammalian target of rapamycin; FoxO1: Forkhead box protein O1. This figure was created with Biorender at www.biorender.com (accessed on 21 August 2023).

**Table 2 ijms-24-16523-t002:** Clinical trials of cannabinoids.

Study Type	Patients (Sample Size)	Treatment	Outcome	Study Results
Observational study (Maghfour, J. et al.) [14]	Atopic dermatitis (*n* = 20)	Topical application of CBD gel	Patient-oriented eczema measure score (POEM) and Quality of Life Hand Eczema Questionnaire (QOLHEQ).	Significant reduction of POEM (*p* < 0.0007) and QOLHEQ (*p* < 0.004)
Retrospective study (Palmieri, B. et al.) [15]	Psoriasis (*n* = 5) Atopic dermatitis (*n* = 5) Scars (*n* = 10)	Topical CBD-enriched ointment twice daily for three months	Skin hydration (TEWL) assessed using a DermaLab^®^ device Elasticity assessed using an ElastiMeter^®^ device Photographic assessment	Significant reduction in TEWL (*p* < 0.001) Significant improvement of elasticity (*p* < 0.001) Significant improvement in symptoms
Randomized, single-blinded, crossover study (Callaway, J. et al.) [16]	Atopic dermatitis (*n* = 20)	Dietary hemp oil and olive oil for 20 weeks	Fatty acid profiles Skin dryness, itchiness, and use of dermal medications, assessed by patient questionnaire Skin transepidermal water loss (TEWL)	Increased levels of essential fatty acids after hemp oil Improvement in clinical symptoms (*p* < 0.05) Intra-group TEWL values decreased (*p* = 0.074)
Randomized, double-blind, controlled trial (Yuan, C. et al.) [29]	Asteatotic eczema (*n* = 60)	Emollient creams containing PEA (N-acylethanolamine) or NEA (N-acetyl ethanolamine) for 28 days	Clinical assessment using Eczema Area and Severity Index Skin surface hydration assessed using Corneometer CM820^®^	Decreased skin erythema, scaling, dryness, and itching (*p* < 0.05) Skin surface hydration was increased (*p* < 0.05)
Split-body clinical study (Del Rosso, J.Q. et al.) [30]	Atopic dermatitis (*n* = 43)	Combination of topical steroid and PEA compared with the sides treated with only topical steroid	Clearance rate	Significant improvement in clearance rate of symptoms
Observational, prospective cohort study (Eberlein, B. et al.) [31]	Atopic dermatitis (*n* = 2456)	Topical application of PEA for 4–6 weeks	Objective symptoms such as dryness, excoriation, lichenification, scaling, erythema, pruritis	Significant improvement with combined score reduction of 58.6% in the entire population (*p* < 0.001)
Observational study (Pulvirenti, N. et al.) [32]	Atopic dermatitis (*n* = 20)	Topical emulsion containing adelmidrol 2% twice daily for 4 weeks	Clinical symptoms	Complete resolution with no side effects in 80% of patients
Observational, non-blinded, prospective cohort study (Szepietowski, J.C. et al.) [43]	Uremic patients (*n* = 21)	Topical treatment of PEA and anandamide twice daily for 3 weeks	Pruritis and scales assessed by questionnaire	Significant reduction in both pruritis and scales (*p* < 0.0001)
Observational study (Schräder, N.H.B. et al.) [47]	Patients with pruritis (*n* = 22)	Emollient cream containing PEA	Pruritis	Reduction in itch (86.4%)
Randomized, single-blinded study (Visse, K. et al.) [46]	Patients with pruritic dry skin (*n* = 100)	PEA-containing lotion	Pruritis intensity assessed by visual analogue scale	No significant difference between the PEA-containing lotion and control groups
Randomized, placebo-controlled study (Puaratanaarunkon, T. et al.) [62]	Plaque-type psoriasis (*n* = 51)	2.5% CBD ointment twice daily for 12 weeks	Psoriasis severity index score (PASI)	Significant reduction of PASI (*p* = 0.026)
Single-blinded study (Ali, A. et al.) [68]	Healthy patients (*n* = 11)	3% cannabis-seed-extract cream twice daily for 12 weeks	Sebum and erythema content assessed using Sebumeter and Mexameter	Significant reduction of sebum production and erythema (*p* < 0.05)
Randomized controlled trial (Zaher, H. et al.) [77]	Alopecia areata (*n* = 30)	Bimatoprost 0.03% solution twice daily for 3 months compared with mometasone furoate 0.1% cream	Severity of Alopecia Tool	Significant improvement of hair-regrowth (*p* = 0.001)

## Data Availability

Not applicable.

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
