# Peer review of "Cannabinoids and Their Receptors in Skin Diseases"

_ijms, 2023, doi:10.3390/ijms242216523_

Round 1
Reviewer 1 Report
Comments and Suggestions for Authors
First, a general overview of the endocannabinoid system must be included before going to the different diseases.
There are many articles referenced, but I find difficult to find a proper guide through the manuscript. Most of the articles are basic science, animal model, and thus, the real effect on patients with the problems, would be not so clear. I would recommend to shorten the text, to add a table compilating main evidences and separating in preclinical/clinical studies, to give an opinion taking all in consideration in the text.
To my eyes, this problem of just compiling information, goes directly linked to conclusion, as the authors explain all the gaps to fill in the knowlegde on how they are involved in pathogenesis and how much can the disease be changed in our patients.
After the paragraphs related to the diseases, where a table of evidences compiling the molecule, concentration, type of study, results... and a written conclusion from the authors, I would add another one regarding the products that can be found and have studied and a brief commentary about the variability of use due to law in cosmetics, educational beliefs of population...
Comments on the Quality of English LanguagePruritis appears several times both in text and graph. Some other typos can be corrected.
Reviewer 2 Report
Comments and Suggestions for Authors
In the review titled "The cannabinoids and their receptors in skin diseases ", the authors summarized updated studies and reports of cannabinoids in dermatologic therapy.
The following issues have been identified during my review:
Inadequate Introduction: The introduction section is notably brief and does not provide a comprehensive overview of cannabinoids. To enhance the article's quality, the authors should consider expanding the introduction to provide a more thorough description of cannabinoids, their significance, and relevance to the research. In addition, it is recommended to include structural details of the primary cannabinoids discussed (THC, CBD, PEA). Providing this information is crucial for a comprehensive understanding of the research.
Lack of Explanations, Insufficient Detail on Referenced Studies: The article lacks sufficient explanations for key concepts and findings. It is essential to provide detailed explanations to ensure that readers can comprehend the research's significance and methodology. Additionally, it is recommended that the authors provide more extensive details on the content of the studies referenced, rather than solely focusing on summarizing their conclusions. This additional depth will contribute to a more comprehensive understanding of the cited literature and its relevance to the current research.
Inadequate Figure Captions: The captions of the figures do not correspond accurately to the content of the figures themselves. This inconsistency creates confusion and makes it difficult for readers to understand the visual data presented.
Unreferenced Citations: Some references cited in the text are not appropriately cited or mentioned in the reference list. Ensuring accurate and complete referencing is essential for academic rigor and integrity.
Inadequate Conclusion: The conclusion section is overly brief and does not effectively highlight the importance of conducting in vitro mechanistic studies to elucidate the underlying pharmacological mechanisms of action. A more comprehensive and insightful conclusion is needed.
Language Issues: The manuscript contains numerous grammatical errors, and there are repetitive words throughout the text. It is essential to thoroughly proofread and edit the manuscript to improve the clarity and coherence of the writing.
Given these significant shortcomings, I suggest that the authors carefully address these issues before resubmitting the manuscript for further consideration.
Comments on the Quality of English Language
The manuscript contains numerous grammatical errors, and need to be thoroughly proofread.
Round 2
Reviewer 1 Report
Comments and Suggestions for Authors
Quite complete review.
Comments on the Quality of English Languageok to my eyes. Minor editing may be considered
Author Response
We sincerely thank you for kind reviews and careful reading on our paper. And we appreciate your time and consideration.
We categorized the studies to enhance reader comprehension.
Also, we described the mechanism of action within the manuscript and figure caption more clearly.
We sincerely thank you for careful comment.
Reviewer 2 Report
Comments and Suggestions for Authors
Significant work has been invested in the revision of the manuscript. The enhancements made to the language, particularly in English, along with the inclusion of two tables, have undeniably elevated the paper's quality. Nevertheless, several remaining issues must be resolved before the article can be deemed suitable for publication.
Structural Organization: The revised manuscript stills exhibiting a lack of clear and coherent structural organization. To enhance reader comprehension, it is imperative to establish distinct divisions between the various studies referenced in the paper. I recommend categorizing these studies into in vitro, preclinical, and clinical trials, allowing for a more structured presentation of their respective findings. Additionally, it is vital to provide a comprehensive description of the mechanism of action of cannabinoids within the manuscript.
Figure Captions: The captions for the figures should be improved to better match the content of the figures. They should be informative and provide a clear understanding of the data presented. This will enhance the reader's comprehension of the figures and their relevance to the article.
Addressing these issues is crucial to ensuring the article meets the required standards for publication. Once these improvements are made, the paper will be in a much stronger position for consideration.
Author Response
We sincerely thank you for kind reviews and careful reading on our paper. And we appreciate your time and consideration.
As you mentioned, we categorized the studies to enhance reader comprehension.
Also, we described the mechanism of action within the manuscript and figure caption more clearly.
We sincerely thank you for careful comment.
Round 3
Reviewer 2 Report
Comments and Suggestions for Authors
Significant improvement has been made.
Comments on the Quality of English LanguageMinor editing of English language required.